# Hsa-mir-3163 and CCNB1 may be potential biomarkers and therapeutic targets for androgen receptor positive triple-negative breast cancer

Pengjun Qiu[☉‡], Qiaonan Guo[☉‡], Qingzhi Yao[‡], Jianpeng Chen[‡], Jianqing Lin[iD]*

Thyroid & Breast Surgery, The Second Affiliated Hospital of Fujian Medical University, Quanzhou, China

☉ These authors contributed equally to this work.
‡ These authors also contributed equally to this work and also share first authorship on this work
* ljq13905977336@163.com

**Data Availability Statement:** The the Four gene expression microarray data sets presented in this study can be found in the GEO database (https://www.ncbi.nlm.nih.gov/geo) at: GSE42568,

## Abstract

Breast cancer (BC) is the most common malignancy in female, but the role of androgen receptor (AR) in triple-negative breast cancer (TNBC) is still unclear. This study aimed to exam the performance of innovative biomarkers for AR positive TNBC in diagnosis and therapies. Four datasets (GSE42568, GSE45827, GSE54002 and GSE76124) were analyzed by bioinformatic methods and the differential expression genes (DEGs) between the AR positive TNBC tissues and normal tissues were firstly identified by limma package and Venn diagrams. Next, Gene Ontologies (GO) and Kyoto Encyclopedia of Genes and Genomes (KEGG) analysis were performed to explore the relationship between these DEGs. Then, the Protein-protein interaction (PPI) network was constructed. CytoHubba and bioinformatic approaches including Molecular Complex Detection (MCODE), Gene Expression Profiling Interactive Analysis (GEPIA), the Kaplan–Meier (KM) plotter and The Human Pro-tein Atlas (THPA) were used to identify the hub genes. Lastly, a miRNA-hub-gene regulatory axis was constructed by use of Target Scan database and ENCORI database. As a result, a total of 390 common DEGs were identified, including 250 up-regulated and 140 down-regulated. GO and KEGG enrichment analysis showed that the up-regulated DEGs were mostly enriched in the cell division, mitotic nuclear division, nucleosome, midbody, protein heterodimerization activity, cadherin binding involved in cell−cell adhesion, systemic lupus erythematosus and alcoholism, while the down-regulated DEGs were mainly enriched in carbohydrate metabolic process, extracellular space, extracellular region, zinc ion binding and microRNAs in cancer. Then, 13 hub genes (CCNB2, FOXM1, HMMR, MAD2L1, RRM2, TPX2, TYMS, CEP55, AURKA, CCNB1, CDK1, TOP2A, PBK) were selected. The survival analysis revealed that only CCNB1 was associated with significantly poor survival (P <0.05) in TNBC patients. Finally, we found that hsa-miR-3163 took part in the regulation of CCNB1 and constructed a potential hsa-miR-3163-CCNB1 regulatory axis. The results of current study suggest that CCNB1 and hsa-miR-3163 may serve as highly potential prognostic markers and therapeutic targets for AR positive TNBC. Our findings may make contributions to the diagnosis and therapies of AR positive TNBC.

GSE45827, GSE54002 and GSE76124. This information is also found in the "Materials and methods" sections of this article.

**Funding:** The author(s) received no specific funding for this work.

**Competing interests:** The authors have declared that no competing interests exist.

## Introduction

According to the latest cancer statistic, breast cancer is the most common malignant tumor in females, being the most important cause of cancer death in many countries [1–3]. TNBC is a special subtype of breast cancer defined as short of expression of the estrogen receptor (ER), progesterone receptor (PR) and human epidermal growth factor receptor 2 (HER2) [4]. Furthermore, TNBC has been subdivided into four categories: luminal androgen receptor (LAR), mesenchymal (MES), basal-like immunosuppressed (BLIS), and basal-like immune activated (BLIA) [5]. In addition, TNBC accounted for approx. 10–15% of the breast cancer with high incidence of metastatic occurrence and poor overall survival [4]. Despite the several standard chemotherapies and immunotherapies, TNBC was still lack of appropriate targets for treatment.

AR is a transcription factor widely involved in the development of breast, and its importance is self-evident [6]. AR was expressed in approx. 10–43% of TNBC, being involved in the invasion and metastasis of malignant tumor [7, 8]. Nevertheless, the function of AR in TNBC was still unclear. In some studies, the expression of AR was associated with favorable outcomes, but in others, the opposite results had been achieved [9–14]. Therefore, it is urgent to understand the molecular mechanisms of AR positive TNBC progression to identify the potential biomarkers and therapeutic targets.

In current study, we collected microarray data of GSE42568, GSE45827, GSE54002 and GSE76124 datasets from GEO. Limma package was used to identify the DEGs between the AR positive TNBC tissue and normal tissue. GO and KEGG pathway analyses were performed to clarify the potential functions and pathways of DEGs. The PPI network was created by the Search Tool for the Retrieval of Interacting Genes (STRING) database and visualized by Cystoscape software. Followed by the core module was constructed by MCODE and the hub genes were selected by the topological analysis methods. The expression levels of hub genes between TNBC tissues and normal tissues were validated by GEPIA (P <0.05). The online database Kaplan-Meier plotter was applied to the hub genes for prognostic evaluation. Subsequently, the CCNB1 was identified as the key gene in AR positive TNBC. Target miRNAs of CCNB1 were predicted by from the intersection of ENCORI and TargetScan, and the correlation between CCNB1 and the miRNAs were verified. Finally, a regulatory miRNA of CCNB1 was identified, has-miR-3163, which was related to the molecular mechanisms of the AR positive TNBC. The result may further provide a new strategy for the diagnosis and treatment of AR positive TNBC.

## Materials and methods

### Microarray data

Four gene expression microarray datasets (GSE42568, GSE45827, GSE54002 and GSE76124) were downloaded from the GEO (https://www.ncbi.nlm.nih.gov/geo) database. All the datasets were processed on the GPL570 platform (Affymetrix Human Genome U133 Plus 2.0 Array). The normal breast tissue samples were selected from GSE42568, GSE45827 and GSE54002, and the AR positive TNBC samples were selected from GSE76124. GSE42568, GSE45827 and GSE54002 contained 17, 11, and 16 normal breast tissue samples, respectively. GSE76124 contained 37 AR positive TNBC samples.

### Data preprocessing

The regression calculation was performed to evaluate the datasets by FITPLM function in the AFFYPLM package. Followed by, the quality of data was evaluated by the weight map, residual

symbol map, relative logarithmic expression map and RNA degradation map. Next, these data-sets were read by RMA method, and the missing values were added by KNN method [15]. Finally, the probe ID was replaced with the corresponding gene symbol by the GPL570 plat-form annotation file.

## Identification of DEGs

The DEGs between AR positive TNBC samples (GSE76124) and normal breast tissue samples (GSE42568, GSE45827 and GSE54002) were analyzed by limma package, respectively. The genes that met cutoff criteria of |log2fold change (FC)|> 1 and p-value < 0.05 were considered as DEGs. Subsequently, the overlapping DEGs were screened out via drawing Venn diagrams for subsequent function analysis.

## GO and KEGG pathway enrichment analysis for DEGs

GO and KEGG analysis included functional annotation, enrichment analysis and pathways analysis for genes. GO analysis contained 3 categories, biological processes (BPs), cellular com-ponents (CCs) and molecular functions (MFs) [16, 17]. Online database DAVID (https://david.ncifcrf.gov/, version 6.8) was employed to perform GO and KEGG pathway analysis for the up-regulated and down-regulated DEGs, respectively (p < 0.05) [18].

## PPI network construction and analysis

The PPI network was constructed for DEGs by the STRING database (https://string-db.org/, version 11.0) with a combined interaction score > 0.4, displayed by Cytoscape (version 3.8.2) [19, 20]. Based on the network, the cytoHubba was used to search for the hub genes [21]. Sub-sequently, the MCODE app in Cytoscape was adopted to check these genes (degree cut-off = 2, node score cut-off = 0.2, k-score = 2, and max Depth = 100).

## Survival analysis and validation of the hub genes

Online database KM plotter(http://kmplot.com/analysis/) was adopted to evaluate the prog-nostic value of the identified hub genes [22]. In our research, the parameters were set as fol-lows: (1) the negative expression of ER, PR and HER-2; (2) selected only JetSet best probe set; (3) excluded biased arrays; (4) survival: RFS. The mRNA expression levels of the hub genes between normal breast samples and BC samples were verified by the GEPIA(http://gepia.cancer-pku.cn/) platform(|log2FC| cut-off>1 and p-value cut-off< 0.01) [23]. THAP database (https://www.proteinatlas.org/) was applied to validate the expression levels of CCNB1 in nor-mal breast tissues and breast tumor tissues [24].

## Identification of candidate miRNAs

ENCORI and TargetScan databases were employed to predict the target miRNAs of CCNB1. In our study, the parameters of ENCORI were set as follows: (1) CLIP data: high stringency (≥3); (2) Degradome data: with or without data; (3) Pan-cancer: 1 cancer type. TargetScan (http://www.targetscan.org/vert_72/; version 7.2) was a widely used database to predict miR-NAs [25]. The candidate miRNAs of CCNB1 were selected through the intersection of ENCORI and TargetScan.

## Identification of the miRNA-mRNA regulation axis

ENCORI was used to verify the correlation between CCNB1 and the candidate miRNAs. It was also used to compare the expression levels of candidate miRNAs between BC samples and

normal tissue samples. The overexpression of CCNB1 was associated with unfavorable prognosis in AR positive TNBC. Accordingly, we hypothesized that the miRNAs regulating CCNB1 was associated with a favorable prognosis. Subsequently, the KM plotter was adopted to assess the prognosis of the candidate miRNAs (the dataset: TCGA, molecular subtype: TNBC). Eventually, the miRNA-mRNA regulation axis was displayed by Cytoscape.

### Drug sensitivity analysis

The NCI-60 database, containing data from 60 cancer cell lines, was analyzed by CELLMINER website (https://discover.nci.nih.gov/cellminer/). The expression status of target genes and z-score for cell sensitivity data (GI50) was downloaded from the website and assessed through Pearson correlation analysis to determine the correlation between target gene expression and drug sensitivity.

## Results

### Data quality assessment

All the points were distributed uniformly in the weight and residual symbol maps. In the relative logarithmic expression map, all samples were close to the zero point without outliers. In the RNA degradation diagram, the 5′-terminal was lower than the 3′-terminal and the slope was suitable [15] (S1 Fig). The results indicated that these data were suitable for further analysis.

### Identification of DEGs

Our study included 44 normal breast tissues and 37 AR positive TNBC tissues, the normal breast tissue datasets (GSE42568, GSE45827 and GSE54002) and AR positive TNBC tissues dataset (GSE76124) were analyzed by R software to obtain DEGs, respectively. GSE76124 vs GSE42568, GSE76124 vs GSE45827 and GSE76124 vs GSE54002 were analyzed, resulting in 2896 (1771 up-regulated and 1125 down-regulated), 3360 (2091 up-regulated and 1269 down-regulated) and 3158 (1166 up-regulated and 1992 down-regulated) DEGs, respectively (Fig 1). Subsequently, the overlapping DEGs were identified via the Venn diagrams. Compared with normal breast tissues, 390 common DEGs were discovered in AR positive TNBC tissues, of

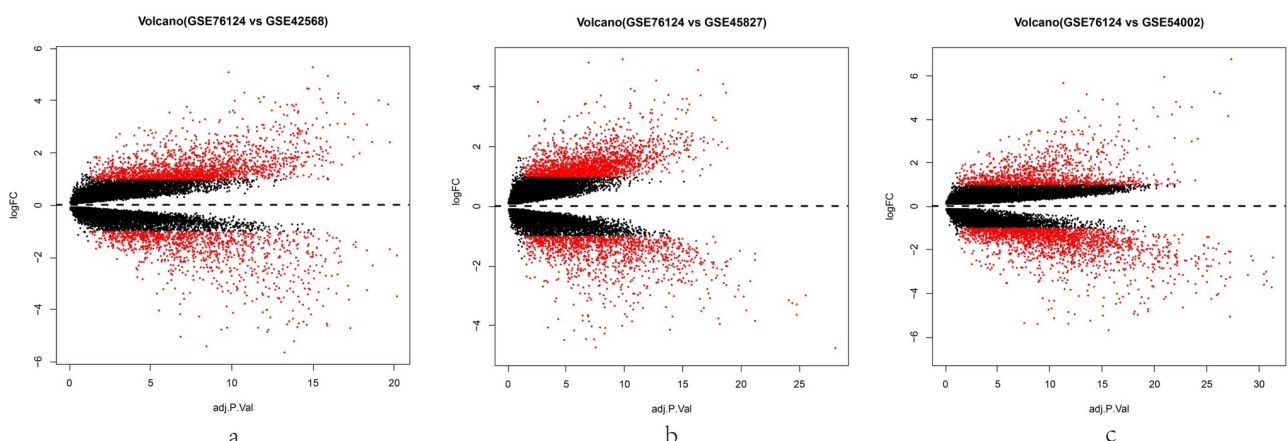

**Fig 1. Volcano map of DEGs, red color represents of the up-regulated and down-regulated genes.** (a) GSE76124 VS GSE42568, (b) GSE76124 VS GSE45827, (c) GSE76124 VS GSE54002.

which 250 were up-regulated (P < 0.05, log2FC > 1) and 140 were down-regulated (P < 0.05, log2FC <-1) (Fig 2).

## GO and KEGG pathway enrichment analysis

The GO and KEGG pathway analysis were performed to explore the biological functions for the 390 overlapping DEGs by means of DAVID software. The top five GO terms in the BP, CC and MF categories are shown in Fig 3. The upregulated DEGs were mostly enriched in cell division and mitotic nuclear division among the BP categories, the nucleosome among the CC categories, and protein heterodimerization activity and cadherin binding involved in cell–cell adhesion among the MF categories. The downregulated DEGs were mainly associated with carbohydrate metabolic process among the BP categories, extracellular space and extracellular region among the CC categories, and zinc ion binding among the MF categories. In the KEGG pathway analysis, the upregulated DEGs were mostly enriched in the systemic lupus erythematosus and alcoholism, and the downregulated DEGs were mostly enriched in MicroRNAs in cancer (Fig 3).

## PPI network construction and analysis

The STRING database was adopted to construct the PPI network for the 390 overlapping DEGs, displayed in Cytoscape. The result revealed that the PPI network contained 287 nodes and 3420 edges (Fig 4a). The most significant module was identified by the MCODE based on the PPI network. The result showed that the major module contained 70 central nodes and 2271 edges, and all of which were up-regulated genes. The topological analysis methods in cytoHubba were adopted to select the hub genes. As a result, CCNB2, FOXM1, HMMR, MAD2L1, RRM2, TPX2, TYMS, CEP55, AURKA, CCNB1, CDK1, TOP2A, PBK were identified as hub genes which were in the intersection of at least five methods. In addition, all the 13 hub genes were contained in the most meaningful module (Fig 4b).

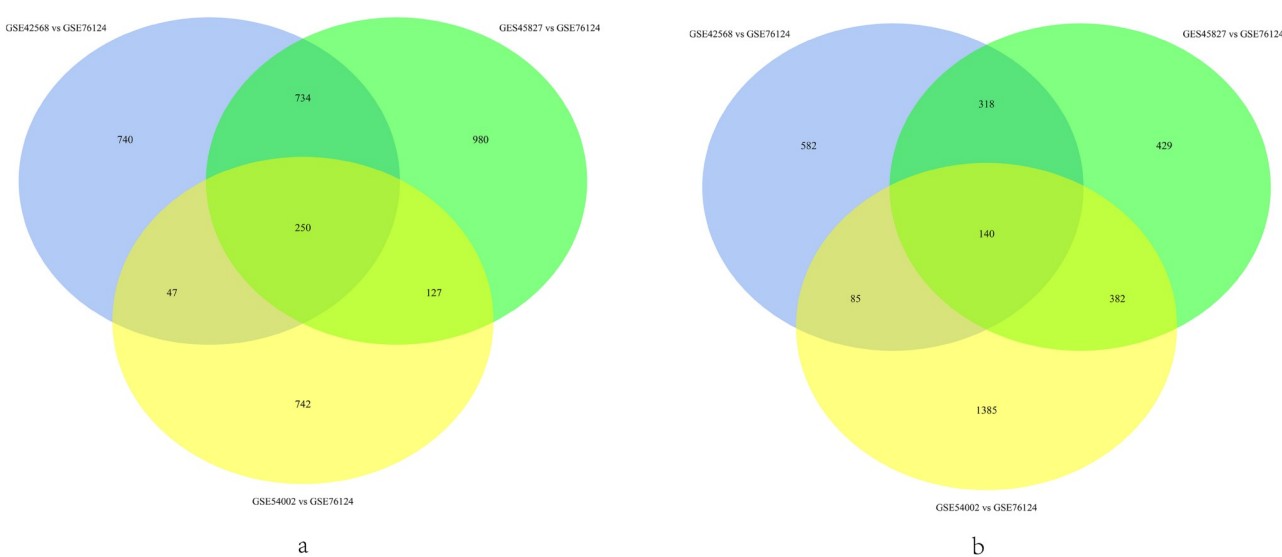

**Fig 2. Venn diagram of DEGs in GSE76124 VS GSE42568, GSE76124 VS GSE45827 and GSE76124 VS GSE54002.** (a) the overlapped 250 up-regulated DEGs (P < 0.05, log2FC > 1), (b) the overlapped 140 down-regulated (P < 0.05, log2FC <-1).

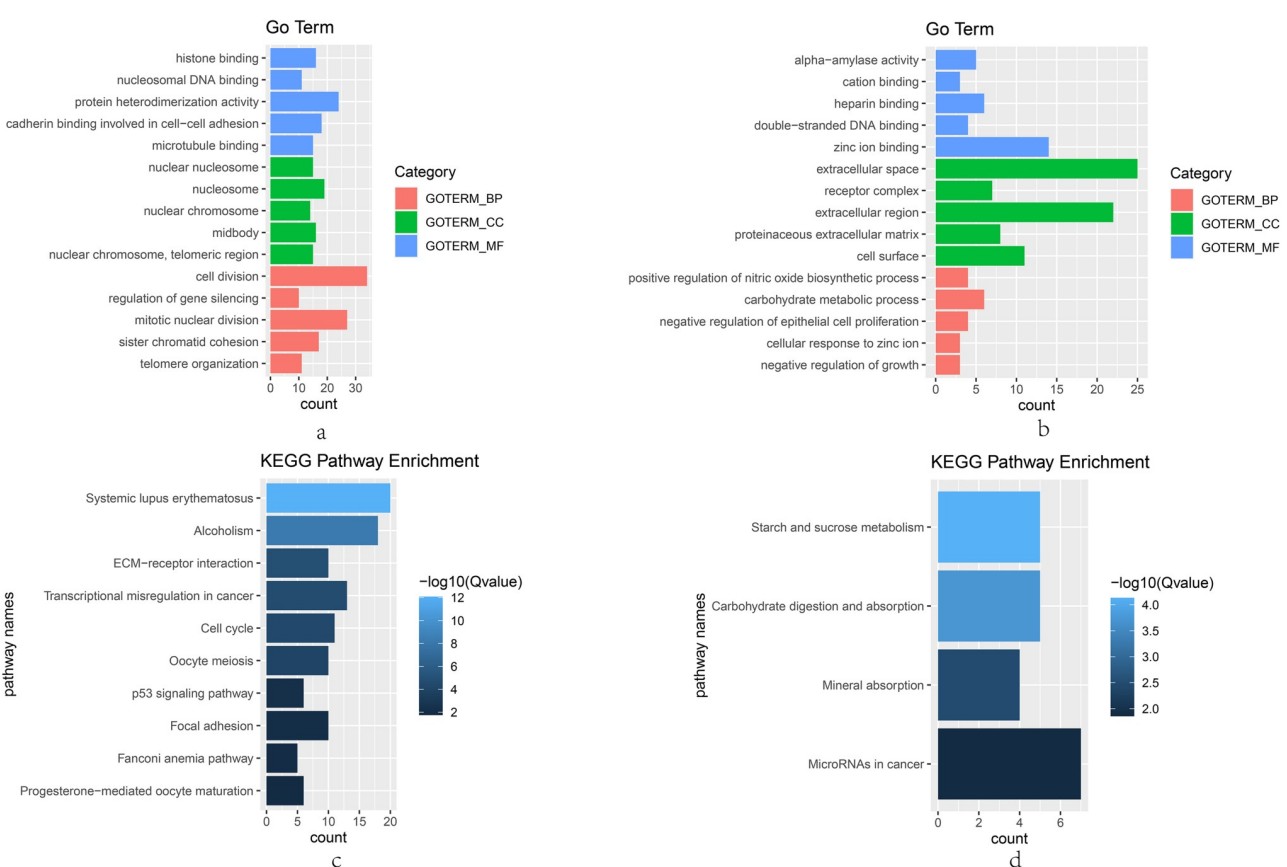

**Fig 3. GO and KEGG pathway enrichment analysis for DEGs.** (a) up-regulated DEGs enrichment in GO, (b) down-regulated DEGs enrichment in GO, (c) up-regulated DEGs enrichment in KEGG pathway, (d) down-regulated DEGs enrichment in KEGG pathway.

## Expression and survival analysis for hub genes

The expression levels of the 13 hub genes were examined by GEPIA database (|log2FC| cut-off value = 1 and p value cut-off value = 0.01). Fig 5 indicated that the expression levels of the 13 hub genes were significantly higher in breast cancer tissues than in normal breast tissues (P < 0.05). Subsequently, the KM plotter was adopted to the survival analysis for-mentioned 12 hub genes (CDK1 data was not found), resulting that only the CCNB1 was associated with statistical poor survival (P<0.05) in TNBC (Fig 6). Eventually, THAP data-base was employed to explore the relative expression of CCNB1 in BC. Immunohistochemis-try (IHC) in THPA database verified that CCNB1 was upregulated in breast cancer tumor tissues (Fig 7).

## Identification of candidate miRNAs

MiRNAs are small regulatory molecules (a short non-coding RNAs) which own vital biological functions [26]. A total of 42 miRNAs regulating CCNB1 were predicted by the ENCORI data-base. 698 miRNAs regulating CCNB1 were predicted by the TargetScan database, including 6 conserved sites and 692 poorly conserved sites. Eventually, 24 overlapping miRNAs were selected via the Venn diagrams (Figs 8 and 9).

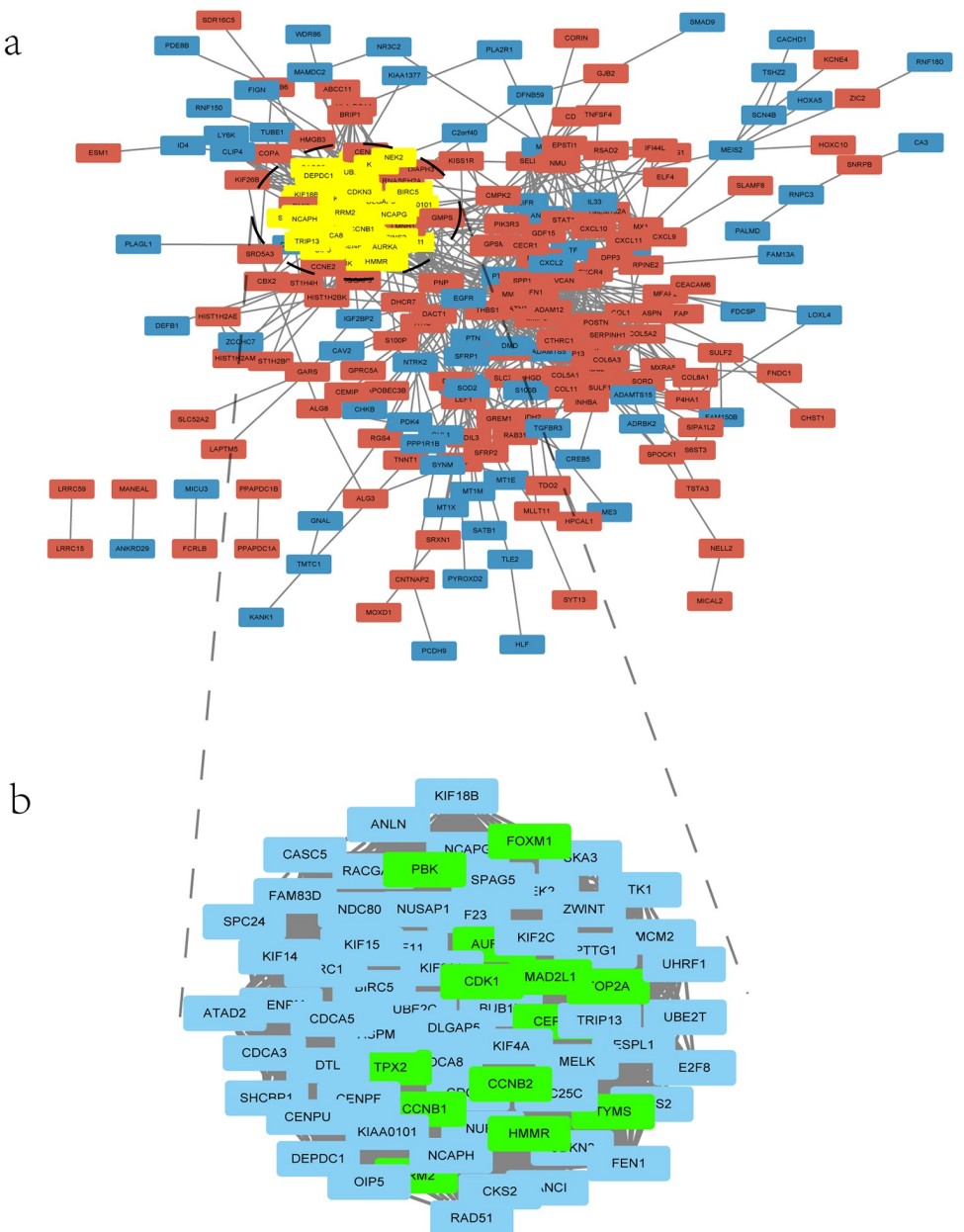

**Fig 4. The PPI network of DEGs and the highest score clustering module of the PPI network.** (a) the PPI network of DEGs, red color indicates up-regulated DEGs, blue color represents downregulated DEGs and yellow color indicates the highest score clustering module generated by MCODE. (b) the highest score clustering module of the PPI network was magnified and green color indicates the hub genes of the PPI network.

## MiRNAs expression analysis and survival analysis

The expression levels of target genes are negatively regulated by miRNAs binding to specific target sites via base pairing interactions to induce mRNA silence and degradation [27, 28]. The miRNA is negatively correlated with the mRNA [29]. Therefore, the expression of the candidate miRNAs regulating CCNB1 should be lower in BC, related with poor prognosis.

ENCORI was used to verify the correlation between the CCNB1 and the candidate miR-NAs. Among the 24 miRNA-CCNB1 pairs, 7 negatively correlated pairs were considered

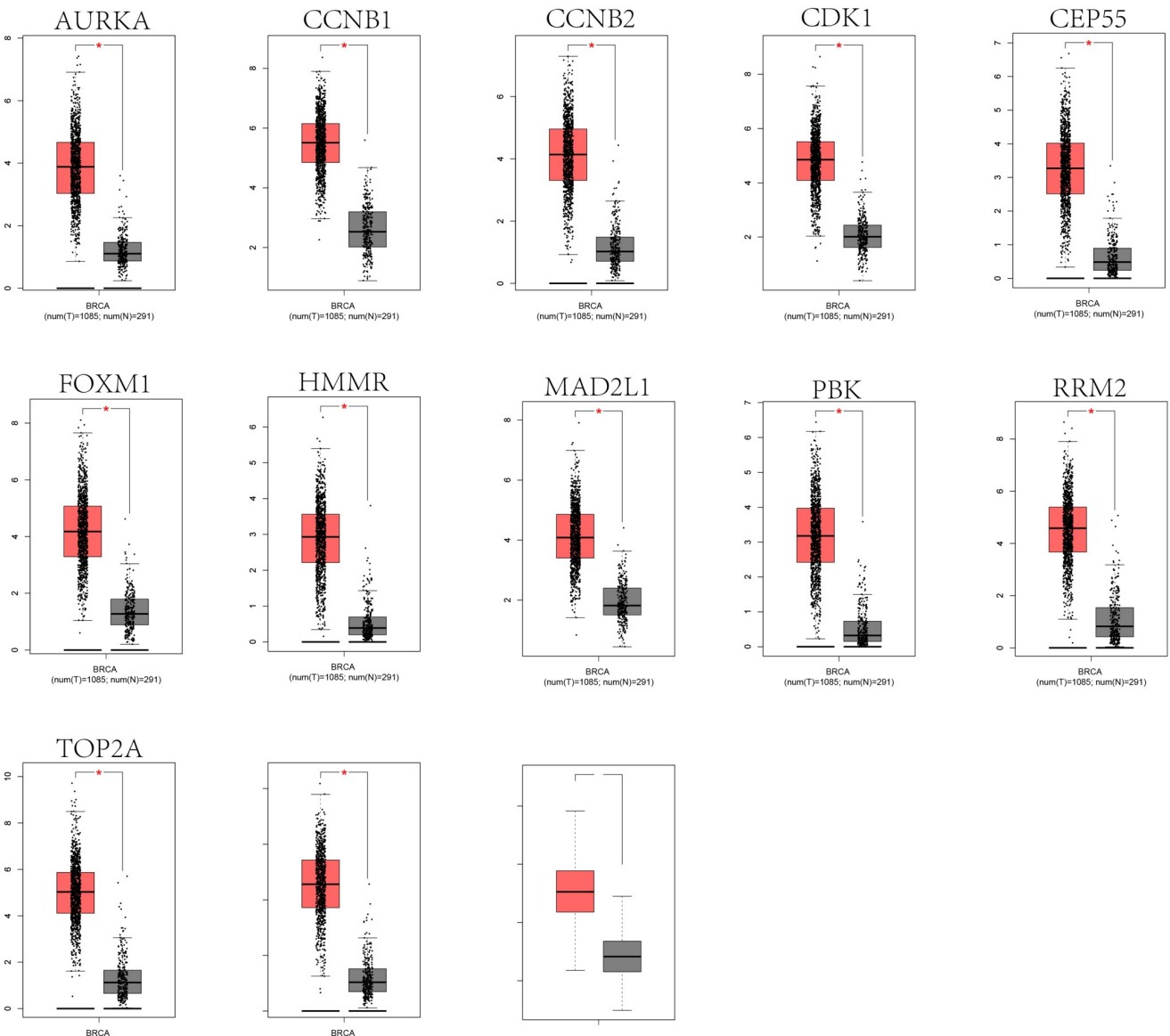

**Fig 5. The expression of 13 hub genes.** All the 13 hub genes highly expressed in breast cancer samples contrasted to normal samples (P < 0.01). Red color represents tumor tissues and grey color represents normal tissues.

statistical (P < 0.05). The KM plotter were applied to survival analysis for the 7 candidate miR-NAs. Fig 10 shows that the low expression of hsa-miR-181C-5p and hsa-miR-3163 are associated with the poor prognosis of TNBC. Finally, the ENCORI pan-cancer analysis was performed to compare the expression levels of hsa-miR-181C-5p and hsa-miR-3163 between breast cancer and normal samples. The result showed that only hsa-miR-3163 was significantly down-regulated in BC samples (Fig 11).

## MiRNA-CCNB1 regulation axis

The hsa-miR-3163 was selected to construct the miRNA-mRNA regulatory axis by bioinformatics analysis, visualized by Cytoscape software (Fig 11c).

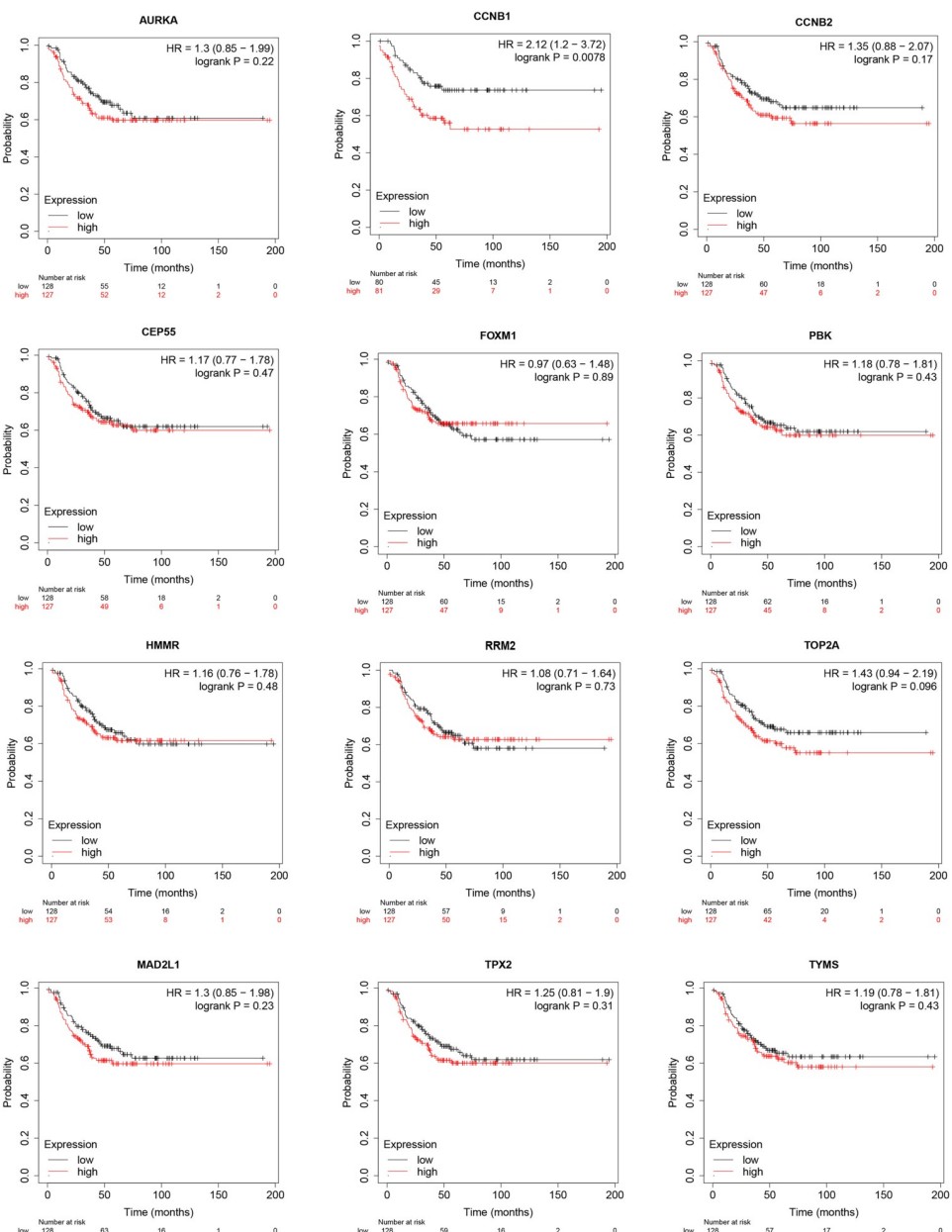

**Fig 6. Kaplan–Meier plotter for the hub genes.** The Kaplan–Meier test p-value < 0.05: CCNB1. The Kaplan–Meier test p-value > 0.05: AURKA, CCNB2, CEP55, FOXM1, PBK, HMMR, RRM2, TOP2A, MAD2L1, TPX2, TYMS.

## Drug sensitivity analysis to CCNB1

The influence of target genes on drug sensitivity was assessed by using CellMiner database, which could facilitate better precision treatment. Drug sensitivity was measured by z-score, and the higher the scores implied that cells were more sensitive to the drug treatment (Fig 12). Notably, CCNB1 was associated with cell resistance to the treatment of denileukin difttitox (ontak). Furthermore, CCNB1was related to the increased sensitivity of cells to thioguanine and allopurinol.

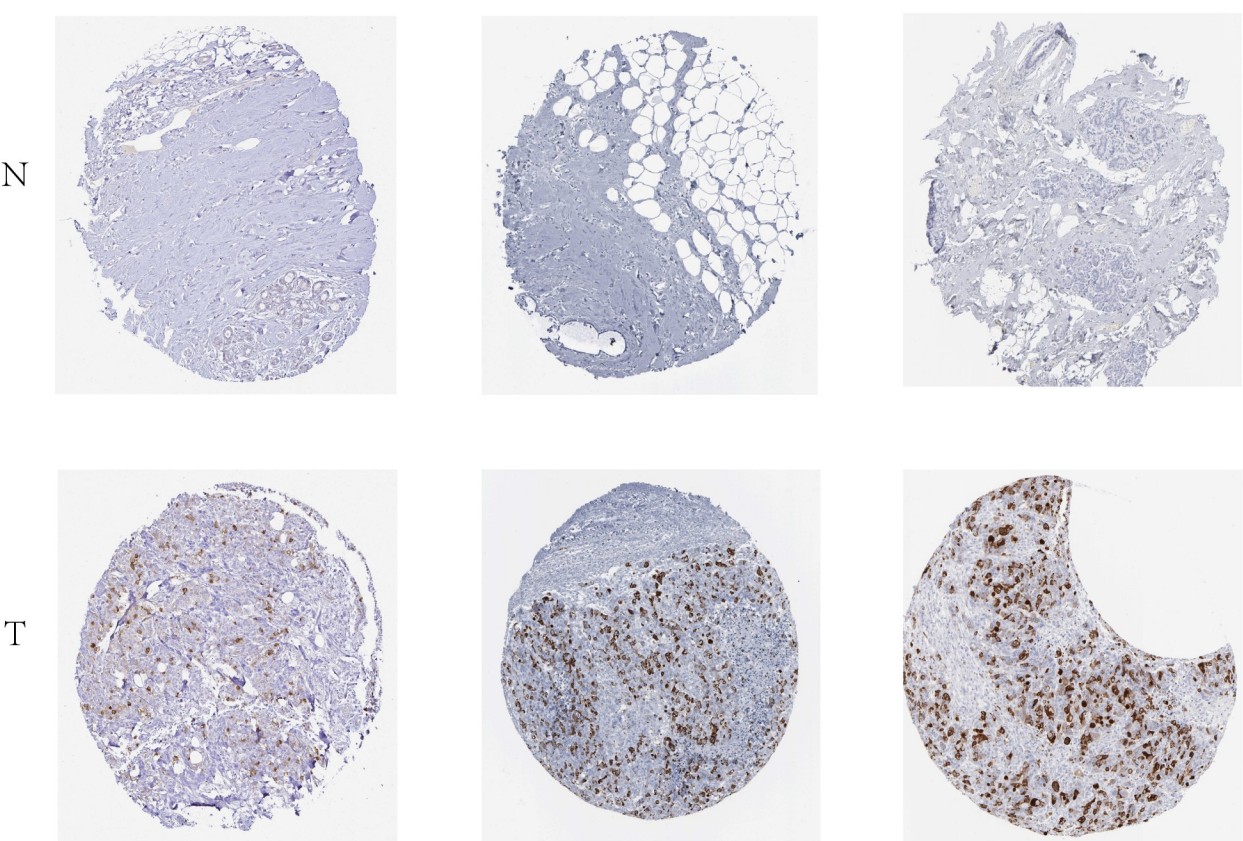

**Fig 7. THPA website analysis of CCNB1 protein in BC.** CCNB1 protein expression in BC specimens and non-cancerous breast tissues via THPA website analysis. Three representative images of BC (https://www.proteinatlas.org/ENSG00000134057-CCNB1/pathology/breast+cancer#img) and non-cancerous breast tissues were presented (https://www.proteinatlas.org/ENSG00000134057-CCNB1/tissue/breast#img). N, normal; T, tumor; BC, breast cancer.

## Discussion

Breast cancer is the most common malignant tumor in female. TNBC is a subtype of BC, lacks the expression of hormone receptors as well as human epidermal growth factor receptor 2 (HER2). Due to the absent of molecular targets, TNBC has no chance of endocrine treatment and HER2 target therapy. Consequently, an increasing number of studies were performed to detect new diagnosis biomarkers and therapeutic targets of TNBC. The role of AR in TNBC remains poorly understood. Therefore, it is significant to explore the mechanisms of AR in TNBC.

At present, the bioinformatic methods were seldom applied to study in AR positive TNBC. In this study, four datasets (GSE42568, GSE45827, GSE54002 and GSE76124 datasets) were analyzed by bioinformatic methods, including 44 normal breast tissues and 37 AR positive TNBC tissues. A total of 390 common DEGs (P < 0.05 and | log2FC |>1) were identified by use of limma package and Venn diagrams, of which 250 were up-regulated (log2FC > 1) and 140 were down-regulated (log2FC < -1). Subsequently, DAVID was used to GO and KEGG pathway enrichment analysis. In the GO enrichment analysis, the DEGs were mainly related to cell division, mitotic nuclear division, nucleosome, protein heterodimerization activity, carbohydrate metabolic process, cadherin binding involved in cell−cell adhesion, extracellular space, and zinc ion binding. In the KEGG pathway analysis, the DEGs were mainly related to

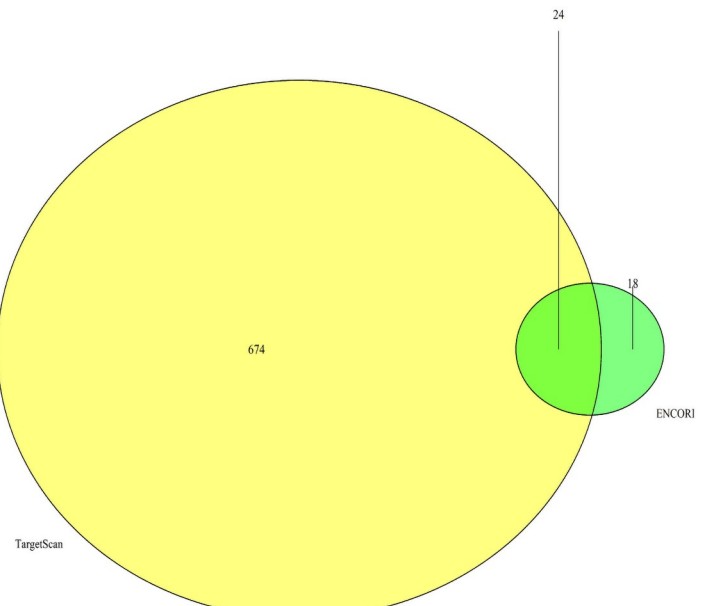

**Fig 8. Validation of the overlapped miRNAs in the two databases (ENCORI and TargetScan) via Draw Venn diagram.**

MicroRNAs in cancer. Afterwards, the PPI network was established, comprehending 287 nodes and 3420 edges. 13 hub genes were selected based on the PPI network. Furthermore, GEPIA showed that the expression levels of the 13 hub genes were higher in breast tumor tissues than in normal tissues (P<0.01). The Kaplan–Meier plotter manifested that only CCNB1 was associated with significantly poorer survival (P<0.05) of TNBC patients. Eventually, IHC in THPA verified that CCNB1 was upregulated in breast cancer tumor tissues. Accordingly, CCNB1 may be a highly potential prognostic marker for AR positive TNBC.

CCNB1 (Cyclin B1) is a member of the cyclin family, which plays an important role in the cell cycle at the G2/M (mitosis) transition [30]. The overexpression of CCNB1 can cause uncontrolled cell growth and carcinogenesis by binding CDK1 [31]. It has been reported that overexpression of CCNB1 was correlated with adverse pathological types and clinical outcomes [32, 33], such as BC [34–36], colon adenocarcinoma [31], non-small cell lung cancer [37] and esophageal squamous cell cancer [38]. A study showed that CCNB1 could be used as a prognostic factor of breast cancer death, with good repeatability [39]. Furthermore, CCNB1 expression levels were significantly increased from benign to malignant breast tissues [36]. Knockdown of CCNB1 could suppress cell proliferation in TNBC [40]. Besides, a study in prostate cancer identified CCNB1 as a bona fide AR target gene in prostate stromal cells. The negative regulation of CCNB1 by AR was mediated through switching between E2F1 and E2F4 on the promoter of CCNB1 [41]. Accordingly, the CCNB1 may be a highly potential biomarker and therapeutic target for AR positive TNBC.

MiRNAs are small non-coding RNA molecules, widely involved in multiple processes of cancer progression including tumor genesis, development, drug resistance and metastasis [42, 43]. MiRNAs negatively regulate target genes by inducing mRNA silencing and degradation. It can be detected on tissue samples and biological fluids including serum, plasma and urine [44]. Therefore, miRNA has been regarded as a promising potential biomarker for cancer diagnosis and treatment. This research aimed to explore the potential miRNAs in AR positive

| NO. | miRNA | Coefficient-R | p-Value |
|---|---|---|---|
| 1 | hsa-miR-139-5p | -0.224 | 8.99E-14 |
| 2 | hsa-miR-410-3p | -0.189 | 3.49E-10 |
| 3 | hsa-miR-379-5p | -0.177 | 4.40E-09 |
| 4 | hsa-miR-181c-5p | -0.135 | 7.83E-06 |
| 5 | hsa-miR-488-3p | -0.098 | 1.19E-03 |
| 6 | hsa-miR-3163 | -0.066 | 2.87E-02 |
| 7 | hsa-miR-181d-5p | -0.065 | 3.28E-02 |
| 8 | hsa-miR-5590-3p | -0.045 | 1.41E-01 |
| 9 | hsa-miR-199a-3p | -0.034 | 2.65E-01 |
| 10 | hsa-miR-199b-3p | -0.033 | 2.79E-01 |
| 11 | hsa-miR-520h | -0.013 | 6.62E-01 |
| 12 | hsa-miR-520g-3p | -0.005 | 8.59E-01 |
| 13 | hsa-miR-144-3p | -0.002 | 9.40E-01 |
| 14 | hsa-miR-513b-5p | 0.01 | 7.30E-01 |
| 15 | hsa-miR-208a-3p | 0.015 | 6.23E-01 |
| 16 | hsa-miR-181a-5p | 0.019 | 5.39E-01 |
| 17 | hsa-miR-208b-3p | 0.033 | 2.77E-01 |
| 18 | hsa-miR-3121-3p | 0.054 | 7.69E-02 |
| 19 | hsa-miR-140-5p | 0.086 | 4.43E-03 |
| 20 | hsa-miR-802 | 0.117 | 1.18E-04 |
| 21 | hsa-miR-181b-5p | 0.138 | 5.37E-06 |
| 22 | hsa-miR-582-3p | 0.171 | 1.43E-08 |
| 23 | hsa-miR-2355-5p | 0.207 | 5.31E-12 |
| 24 | hsa-miR-142-5p | 0.253 | 2.92E-17 |

**Fig 9. Correlation between miRNA-CCNB1 pairs identified by ENCORI database.**

TNBC as biomarkers and therapeutic targets. 24 miRNAs regulating CCNB1 were predicted by TargetScan database and ENCORI database, of which 13 were downregulated in breast cancer. And only 7 of the 13 miRNAs were considered statistically significant (P < 0.05). Afterwards, the KM plotter was adopted to assess the prognostic value of the 7 miRNAs in TNBC. Consequently, the hsa-miR-3163 was identified with good prognostic value in TNBC.

Functionally, hsa-miR-3163 played as a tumor suppressor. A recent study in breast cancer indicated that MiR-513c and miR-3163 were downregulated in tumor tissues, which might serve as tumor suppressors [45]. This is in line with the results of our research. Although there were few studies on the function of miR-3163 in breast cancer, miR-3163 were identified as a tumor suppressor in several kinds of malignant tumors. It has been reported that overexpression of hsa-miR-3163 may enhance the sensitivity of hepatocellular carcinoma cells to molecular targeted agents [46]. In non-small cell lung cancer, hsa-miR-3163 could inhibit tumor cell growth [47]. In retinoblastoma cancer stem cells, hsa-miR-3163 was associated with proliferation, apoptosis and multi-drug resistance [48]. The expression of hsa-miR-3163 has been downregulated in colorectal cancer, whereas the colorectal cancer cell proliferation is suppressed by the overexpression [49, 50]. In the future, further studies in vivo and in vitro should be conducted to identify the functional mechanism of miR-3163 in AR positive breast cancer.

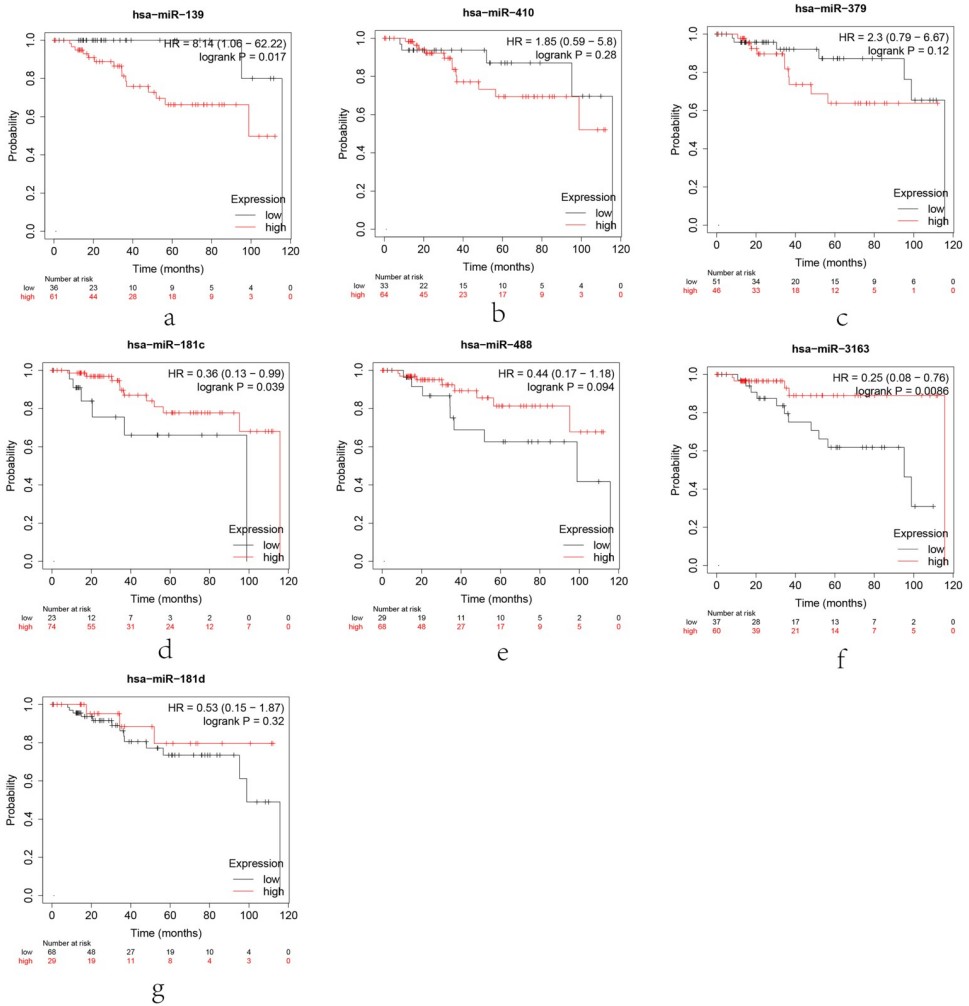

**Fig 10. Kaplan–Meier plotter for miRNAs negatively correlated with CCNB1 expression in TNBC patients.** TNBC patients with low expression of hsa-miR-181C-5p (d) and hsa-miR-3163 (f) had a poor prognosis (P < 0.05).

Taken together, in accord with previous studies, our research indicated that hsa-miR-3163 was related to inhibition of malignant tumors, such as breast cancer. As we know, there were no study investigating the role of hsa-miR-3163 in AR positive TNBC. According to our study, hsa-miR-3163 and CCNB1 may be highly potential biomarkers and therapeutic targets for AR positive TNBC. However, some limitations in our study should be noticed. Firstly, the common data sources from TCGA and GEO database were analyzed only by bioinformatic methods without in vivo and in vitro experiments. Secondly, our study only preliminarily revealed the expression levels of CCNB1 and his-miR-3163 in AR positive TNBC, but seldom involved the functional mechanisms and signaling pathways. Therefore, additional prospective studies are needed to verify the value of CCNB1 and his-miR-3163 in AR positive TNBC and the interaction of them should be further explored by experiment.

## Conclusions

In summary, through bioinformatics analysis, we suggest CCNB1 and hsa-miR-3163 as innovative potential biomarkers for AR positive TNBC. Our findings provide crucial information

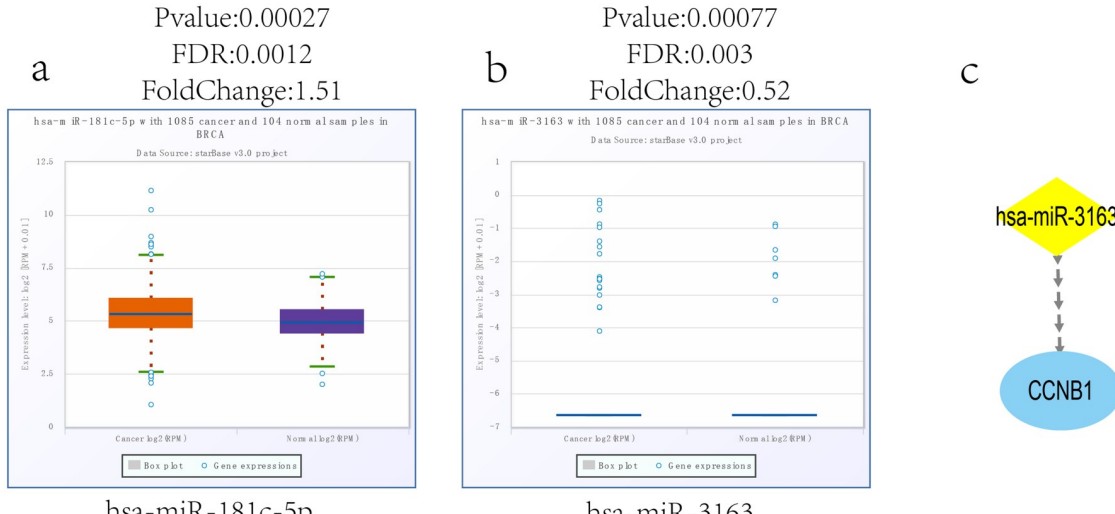

**Fig 11. The expression levels of miRNAs.** Hsa-miR-181C-5p. (a) and hsa-miR-3163 (b) between breast cancer and normal samples. The miRNA-CCNB1 regulation axis (c).

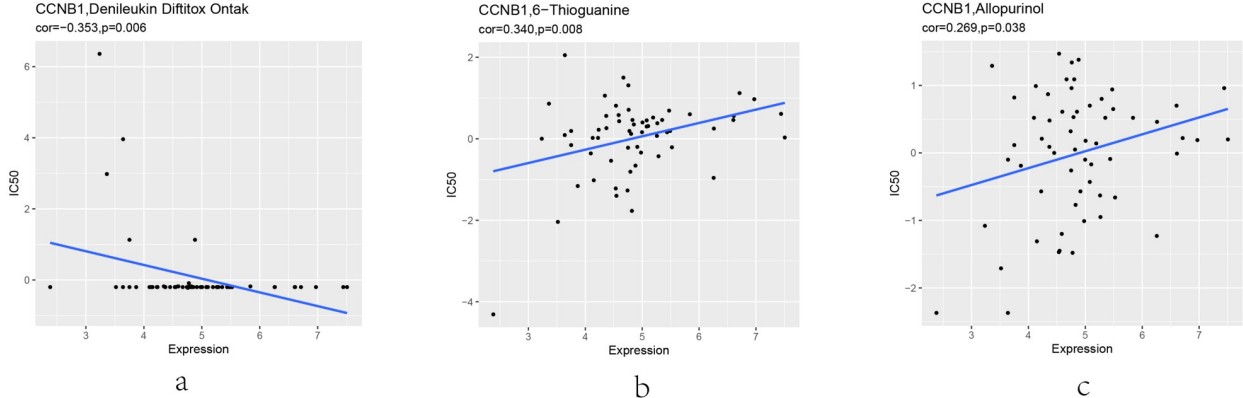

**Fig 12. Drug sensitivity analysis to CCNB1.** The scatter plot indicates the correlation between target gene expression and drug sensitivity (the z-score of the CellMiner interface) for the Pearson correlation test using NCI-60 cell line data. 3 associations are shown, ordered by P-value.

and direction for the elucidation of potential biomarkers and the therapeutic targets of AR positive TNBC. A series of future experiments will be conducted to verify these predictions.

## Supporting information

**S1 Fig. Maps to evaluate the quality of the data.** (a) weight map, (b) residual symbol map, (c) relative logarithmic expression map, (d) RNA degradation map.
(TIF)

## Author Contributions

**Conceptualization:** Pengjun Qiu, Qiaonan Guo, Jianqing Lin.

**Data curation:** Pengjun Qiu, Qiaonan Guo, Qingzhi Yao.

**Formal analysis:** Pengjun Qiu, Qiaonan Guo, Jianpeng Chen, Jianqing Lin.

**Methodology:** Pengjun Qiu, Qiaonan Guo, Jianqing Lin.

**Resources:** Pengjun Qiu.

**Software:** Pengjun Qiu, Qiaonan Guo, Jianpeng Chen.

**Visualization:** Qingzhi Yao.

**Writing – original draft:** Pengjun Qiu.

**Writing – review & editing:** Qiaonan Guo, Qingzhi Yao, Jianpeng Chen, Jianqing Lin.

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
