## [Decision Letter · Decision Letter 0]

18 Aug 2021

PONE-D-21-18875

Hsa-mir-3163-CCNB1 regulatory axis may be potential prognostic marker and therapeutic target for androgen receptor positive triple-negative breast cancer

PLOS ONE

Dear Dr. Jianqing Lin,

Thank you for submitting your manuscript to PLOS ONE. After careful consideration, we feel that it has merit but does not fully meet PLOS ONE’s publication criteria as it currently stands. Therefore, we invite you to submit a revised version of the manuscript that addresses the points raised during the review process.

We look forward to receiving your revised manuscript.

Kind regards,

Suhwan Chang

Academic Editor

PLOS ONE

Journal Requirements:

2. Please modify the title to ensure that it is meeting PLOS’ guidelines (https://journals.plos.org/plosone/s/submission-guidelines#loc-title). In particular, the title should be "specific, descriptive, concise, and comprehensible to readers outside the field" and in this case you should specify that the study was conducted in silico

Reviewers' comments:

Reviewer's Responses to Questions

**Comments to the Author**

1. Is the manuscript technically sound, and do the data support the conclusions?

Reviewer #1: No

Reviewer #2: Yes

2. Has the statistical analysis been performed appropriately and rigorously? 

Reviewer #1: Yes

Reviewer #2: Yes

3. Have the authors made all data underlying the findings in their manuscript fully available?

Reviewer #1: No

Reviewer #2: Yes

4. Is the manuscript presented in an intelligible fashion and written in standard English?

Reviewer #1: No

Reviewer #2: Yes

5. Review Comments to the Author

Reviewer #1: Qiu et al. PONE-D-21-18875 manuscript describes that Hsa-mir-3163 targets on CCNB1, implying a potential marker as the prognostic and therapeutic target for the subtype of AR-TNBC cancer. The topic has a merit for a potential consideration of possible publication. However, the results are ordinary, the discussion is subject to adjustment, and the written English needs to be improved. Therefore, the reviewer would suggest an opportunity of Major Revision for the authors to address the major concern, the major technical questions, and the minor suggestions as follows.

1 The major concern:

Do the authors address possible repeat contents with a recent paper published by another group from the same affiliation of Fujian Medical University (see Ref.22) but has different resulting data?

22.Hong Zhipeng et al., Identification of Seven Cell Cycle-Related Genes with Unfavorable Prognosis and Construction of their TFmiRNA-mRNA regulatory network in Breast Cancer. Journal of Cancer, 2021; 12(3): 740-753. doi: 10.7150/jca.48245. This group is from Fujian Medical University. In this article, GSE42568, GSE45827 and GSE54002 were analyzed by the similar routinely used methods, which resulted in 124 DEGs and led to seven genes (cyclin E2 (CCNE2), cyclin B1 (CCNB1), cyclin B2 (CCNB2), mitotic checkpoint serine/threonine kinase B (BUB1B), dual-specificity protein kinase (TTK), cell division cycle 20 (CDC20), and pituitary tumor transforming gene 1 (PTTG1)) as hub.

In the submitted manuscript, Pengjun Qiu et al. Hsa-mir-3163-CCNB1 regulatory axis may be potential prognostic marker and therapeutic target for androgen receptor positive triple-negative breast cancer. This group is also from Fujian Medical University. Four GEO datasets (GSE42568, GSE45827, GSE54002 and GSE76124) were downloaded from the public GEO depository, and analyzed by the routinely used methods with default parameters. GSE76124 provided 37 AR positive TNBC cancer samples, whereas the rest GSE42568, GSE45827 and GSE54002 datasets provided 17, 11, 16 normal samples (i.e., no cancer). Compared GSE76124 with GSE42568, GSE45827 and GSE54002, respectively, the Limma package generated the differentially expressed genes (DEGs) under the criteria of |log2fold change (FC)|> 1 and p-value < 0.05, including 2896 (1771 up-regulated and 1125 downregulated), 3360 (2091 up-regulated and 1269 down-regulated) and 3158 (1166 upregulated and 1992 down-regulated) DEGs, respectively. These DEGs were the basis of subsequent analyses described in the manuscript. They found 13 hub genes (CCNB2, FOXM1, HMMR, MAD2L1, RRM2, TPX2, TYMS, CEP55, AURKA, CCNB1, CDK1, TOP2A, PBK), and only the CCNB1 was associated with significantly poor survival (P <0.05) in TNBC.

2 The major technical questions:

(1) GSE42568, GSE45827, GSE54002 and GSE76124 were updated by March, 2019 with more than 900 samples. The authors should clearly list out which samples (GSMseries) were used in the analysis, and stated why they were chosen, and which were similar to the Ref.22 paper, which were not, and why. If similar datasets were analyzed, the authors should clear state whether the resulting data were similar or not and why. The ref.22 paper has provided bench-experiments data supporting their findings. The submitted manuscript did not provide such bench-experiments data arguing the discrepancy or similarity of resulting data between the ref.22 and this submission.

(2) MicroRNA-3163 has broad targets across diverse cancers, and CCNB1 has diverse targets across diverse cancers. The authors should discuss the specificity of the two molecules towards the implication of being a potential marker for the defined AR-TNBC cancer. The references below were listed in the submitted manuscript, but did not be well utilized for this purpose.

44. Yang B, Wang C, Xie H, Wang Y, Huang J, Rong Y, et al. MicroRNA-3163 targets ADAM-17 and enhances the sensitivity of hepatocellular carcinoma cells to molecular targeted agents. Cell death & disease. 2019;10(10):784. doi: 10.1038/s41419-019-2023-1. PubMed PMID: 31611551.

45. Su L, Han D, Wu J, Huo X. Skp2 regulates non-small cell lung cancer cell growth by Meg3 and miR-3163. Tumour biology : the journal of the International Society for Oncodevelopmental Biology and Medicine. 2016;37(3):3925-31. doi: 10.1007/s13277-015-4151-2. PubMed PMID: 26482610.

46. Jia M, Wei Z, Liu P, Zhao X. Silencing of ABCG2 by MicroRNA-3163 Inhibits Multidrug Resistance in Retinoblastoma Cancer Stem Cells. Journal of Korean medical science. 2016;31(6):836-42. doi: 10.3346/jkms.2016.31.6.836. PubMed PMID: 27247490.

47. Ren H, Li Z, Tang Z, Li J, Lang X. Long noncoding MAGI2-AS3 promotes colorectal cancer progression through regulating miR-3163/TMEM106B axis. Journal of cellular physiology. 2020;235(5):4824-33. doi: 10.1002/jcp.29360. PubMed PMID: 31709544.

48. Liu D, Zhang H, Cong J, Cui M, Ma M, Zhang F, et al. H3K27 acetylation-induced lncRNA EIF3J-AS1 improved proliferation and impeded apoptosis of colorectal cancer through miR-3163/YAP1 axis. Journal of cellular biochemistry. 2020;121(2):1923-33. doi: 10.1002/jcb.29427.PubMed PMID: 31709617.

3 The minor suggestions:

(1) Please seek for a professional editing service on improving the quality of manuscript so that the science logic and language usage are much better understandable.

(2) Please list out the exact series GSEseries of GEO datasets that were used in the study so that colleagues may reproduce the essential data.

(3) Cutoff non-essential figures (e.g., Figs 1, 2), but clarify essential figures (e.g., Fig.5 is barely readable).

Reviewer #2: The manuscript entitled "Hsa-mir-3163-CCNB1 regulatory axis may be potential prognostic marker and therapeutic target for

androgen receptor positive triple-negative breast cancer" dealt with diagnosis and therapies of AR positive TNBC. The whole manuscript is well documented. Materials and methods are mentioned in detail. The results are clear and well elaborated. Alothough logical discussion may need further improvement. Though this research aricle article is well presented, the voids in the manuscript given below are necessary to be addressed:

1. Please revise the conclusion

2. There are some typing and grammer errors in the manuscript. please revise it carefully throughout the manuscript.

3. Please improve the logical discussion on the results

6. PLOS authors have the option to publish the peer review history of their article (what does this mean?). If published, this will include your full peer review and any attached files.

Reviewer #1: No

Reviewer #2: **Yes: **Muhammad Anwar

---

## [Author Response · Author response to Decision Letter 0]

3 Oct 2021

Response to reviewer 1

Thank you for your precious comments and advice. Those comments are all valuable and very helpful for revising and improving our paper, as well as the important guiding significance to our research. 

1. As for the Ref.22 from Fujian Medical University, Professor Hong Zhipeng and colleagues identified 7 cell cycle-related genes with unfavorable prognosis in breast cancer. The tumor samples and normal samples from GSE42568, GSE45827 and GSE54002 were all analyzed in this article, and 124 DEGs were identified between the breast tumor samples and normal samples. However, we focused on the AR positive triple-negative breast cancer (TNBC), which is a special subtype of breast cancer. Hence, the DEGs and Hub genes were different between these two studies. TNBC is one of the worst prognosis subtypes of breast cancer due to a lack of effective chemotherapy drugs and targeted drugs. Therefore, the study of TNBC is as important as the study in breast cancer. Furthermore, TNBC has been subdivided into four categories: luminal androgen receptor (LAR), mesenchymal (MES), basal-like immunosuppressed (BLIS), and basal-like immune activated (BLIA). Currently, there are seldom studies in the DEGs, tumor progression and treatment of AR positive TNBC. Besides, the tumor samples included in our study were different from these in Professor Hong’s study. In our study, 37 AR positive TNBC samples from GSE76124 and 17, 11 and 16 normal samples from GSE42568, GSE45827 and GSE54002 were included. The normal samples from these two studies were the same, but tumor samples were different. Based on these two reasons, we consider that the different findings from Professor Hong’s team and our team are reasonable and meaningful. The specific samples (GSMseries) were uploaded as additional materials (文件名). Unfortunately, we are currently not in a position to carry out the validation trials about the 13 Hub genes. I am sorry about the poor condition. We hope to have chances to conduct a wet experiment to further validate the results in the future and share them with all of you. Thanks again for your careful review and precious comments.

2. Thank you for your precious comments and advice. The specificity of microRNA-3163 and CCNB1 towards the implication of being a potential marker for AR positive TNBC were reconsidered and discussed. Also, the references have been modified to be more reasonable. The manuscript with tracked changes have been re-submitted. 

3. Thank you for your careful review and precious and advice. We are very sorry for the mistakes in this manuscript and inconvenience they caused in your reading. We have revised our manuscripts according to your advice. The exact series GSEseries of GEO datasets that were used in our study has been list out and uploaded as supplementary file. Figure 1 has been removed from the manuscript and submitted as supplementary material. The Figure 5 as well as the figure legend of it has been modified.

Thank you for reviewing our manuscript again and making valuable suggestions. We hope that the revised manuscript is approved for publication.

 

Response to reviewer 2

Thank you for your summary. We really appreciate your efforts in reviewing our manuscript. We have studied comments carefully and have made correction which we hope to meet with approval. We have modified the conclusion to be more preciseness. The results of the study were revisited carefully. We are very sorry for the mistakes in this manuscript and inconvenience they caused in your reading. We have revised our manuscripts and modified the typing and grammar errors. Thanks again for your effort. We have re-submitted the new manuscript and supplementary file. We hope that the revised manuscript is approved for publication in PLOS ONE.

---

## [Editor Report · Decision Letter 1]

3 Nov 2021

Hsa-mir-3163 and CCNB1 may be potential biomarkers and therapeutic targets for androgen receptor positive triple-negative breast cancer

PONE-D-21-18875R1

Dear Dr. Jianqing Lin,

We’re pleased to inform you that your manuscript has been judged scientifically suitable for publication and will be formally accepted for publication once it meets all outstanding technical requirements.

Kind regards,

Suhwan Chang

Academic Editor

PLOS ONE

Additional Editor Comments (optional):

No further comments
---

## [Editor Report · Acceptance letter]

11 Nov 2021

PONE-D-21-18875R1 

Hsa-mir-3163 and CCNB1 may be potential biomarkers and therapeutic targets for androgen receptor positive triple-negative breast cancer 

Dear Dr. Lin:

I'm pleased to inform you that your manuscript has been deemed suitable for publication in PLOS ONE. Congratulations! Your manuscript is now with our production department. 

Kind regards, 

on behalf of

Dr. Suhwan Chang 

Academic Editor

PLOS ONE